# Fast and Scalable Incomplete Multi-View Clustering with Duality Optimal Graph Filtering

Liang Du
Shanxi University
Taiyuan, China
duliang@sxu.edu.cn

Yukai Shi
Shanxi University
Taiyuan, China
shiyukai7021@163.com

Yan Chen
Taiyuan University of Technology
Taiyuan, China
1272374859@qq.com

Peng Zhou
Anhui University
Hefei, China
zhoupeng@ahu.edu.cn

Yuhua Qian*
Shanxi University
Taiyuan, China
jinchengqyh@126.com

## Abstract

Incomplete Multi-View Clustering (IMVC) is crucial for multi-media data analysis. While graph learning-based IMVC methods have shown promise, they still have limitations. The prevalent first-order affinity graph often misclassifies out-neighborhood intra-cluster and in-neighbor inter-cluster samples, worsened by data incompleteness. These inaccuracies, combined with high computational demands, restrict their suitability for large-scale IMVC tasks. To address these issues, we propose a novel Fast and Scalable IMVC with duality Optimal graph Filtering (FSIMVC-OF). Specifically, we refine the clustering-friendly structure of the bipartite graph by learning an optimal filter within a consensus clustering framework. Instead of learning a sample-side filter, we optimize an anchor-side graph filter and apply it to the anchor side, ensuring computational efficiency with linear complexity, supported by the provable equivalence between these two types of graph filters. We present an alternative optimization algorithm with linear complexity. Extensive experimental analysis demonstrates the superior performance of FSIMVC-OF over current IMVC methods. The codes of this article are released in https://github.com/sroytik/FSIMVC-OF.

## CCS Concepts

• **Information systems** → **Clustering**; • **Computing methodologies** → **Cluster analysis**.

## Keywords

Incomplete multi-view clustering, large-scale clustering, dual graph filtering, optimal graph filtering, bipartite graph

**ACM Reference Format:**
Liang Du, Yukai Shi, Yan Chen, Peng Zhou, and Yuhua Qian. 2024. Fast and Scalable Incomplete Multi-View Clustering with Duality Optimal Graph Filtering. In *Proceedings of the 32nd ACM International Conference on Multimedia (MM '24), October 28-November 1, 2024, Melbourne, VIC, Australia.* ACM, New York, NY, USA, 10 pages. https://doi.org/10.1145/3664647.3681346

*Corresponding author.

## 1 Introduction

Multi-view clustering (MVC) plays a crucial role in unsupervised learning, seamlessly integrating diverse data types like images, audio, and text found in multimedia content [3, 7, 47]. Despite its importance, real-world applications often grapple with incomplete data, compromising the effectiveness of MVC methods that rely on complete datasets. This has spurred increasing attention on IMVC settings [34, 39], tackling the widespread challenge of missing data in open environments. IMVC methods encompass matrix learning-based [13, 14, 17, 20, 21, 23, 24, 53], graph learning-based [16, 18, 22, 36, 37], and deep learning-based techniques [19, 44, 45].

Among them, graph based methods have demonstrated notable potential in tackling data incompleteness in IMVC by harnessing inter-point relationships to bolster clustering efficacy. They construct affinity graphs that reflect the pairwise similarities among data samples, guiding the clustering process. These graphs leverage connectivity and structural insights to infer and fill in missing data effectively. Moreover, these methods enhance the affinity graph by integrating ancillary data or constraints, which improves clustering outcomes. Techniques such as incorporating global insights or domain-specific knowledge result in more robust graphs that are less affected by incomplete data.

While graph learning-based methods have made strides in various applications, they still encounter significant hurdles in large-scale IMVC. Firstly, the accuracy of the first-order affinity graph is inherently limited. The first-order sample-sample or sample-anchor [16, 22, 36] affinity graph is a cornerstone of many graph-based clustering methods, but its accuracy is inherently constrained. This limitation often caused in the misclassification of out-of-neighborhood intra-cluster samples as negatives and in-neighborhood inter-cluster samples as positives [25]. The degradation of the graph is further exacerbated by data incompleteness. Meanwhile, accurately imputing missing data for such inaccurate graphs without bringing additional noise and errors also poses significant difficulty [18]. Secondly, the computational cost becomes prohibitive, especially when performing operations like eigen-decomposition or matrix inversion on large datasets. On the one hand, there is an expectation to carefully explore the underlying clustering-friendly structure from

incomplete information to improve the quality of affinity graphs, even though this inevitably increases computational complexity. On the other hand, there is a necessity to keep computational costs at a moderate or consistent level. In light of these challenges, there is a imperative demand for the development of innovative approaches capable of simultaneously addressing these conflicting objectives, particularly within the context of large-scale IMVC scenarios.

In response to these challenges, we present the Fast and Scalable Incomplete Multi-View Clustering method with duality Optimal graph Filtering (FSIMVC-OF). We begin by constructing view-independent bipartite graphs to capture the first order affinity between samples and anchors. Next, we introduce a novel sample-sample graph filter derived from these bipartite graphs, capable of capturing higher-order interactions between sample-anchor-samples. Crucially, our method dynamically learns the optimal coefficients of different orders during the clustering procedure. We then apply this sample-sample graph filter to the original bipartite graph, enhancing clustering structure clarity through its low-pass properties. To reduce the computational burden of sample-side graph filtering, we rigorously establish the equivalence between sample-sample and anchor-side filtering, simplifying optimization with the linear complexity. Additionally, we propose a unified framework to learn a consensus clustering representation from these filtered signals. We offer a convergence-guaranteed optimization algorithm for practical applicability. With a computational complexity of $O(n)$, our method is ideal for large-scale datasets. Extensive experiments across various benchmarks demonstrate the superiority of FSIMVC-OF over state-of-the-art IMVC methods.

- We propose a novel method to improve the quality of first order bipartite graph by the duality optimal graph filtering. It leverages optimal graph filter learning on the sample-side to encapsulate higher-order interactions, guiding the enhancement of clustering-friendly structures for the first-order bipartite graphs. Meanwhile, it maintains adaptive filter learning on the anchor-side, ensuring computational efficiency with linear complexity, based on the provable equivalence between these two types of graph filters.
- We propose the learning of a unified consensus clustering representation from these advanced graph signals, supported by a fast optimization algorithm guaranteeing convergence, thus setting our approach apart in scalability and efficacy.
- Extensive experiments comparing with ten state-of-the-art IMVC methods on nine datasets show that FSIMVC-OF outperforms other leading IMVC methods, underscoring its effectiveness and superiority in the field.

## 2 Related Work

To tackle the IMVC problem, several approaches have been developed [34, 39]. These approaches can be categorized into three groups based on differences of learning frameworks: matrix learning-based IMVC [13, 14, 17, 20, 21, 23, 24, 53], graph learning-based IMVC [16, 18, 22, 36, 37], and deep learning-based IMVC [19, 44, 45].

Matrix learning-based IMVC methods interpolate missing terms in partial matrices and fall into three sub-categories: (1) kernel learning-based methods [17, 23, 24], which handle incomplete kernels using imputation and kernel-based techniques; (2) subspace

learning based methods [20, 53], which project multi-view data onto low-dimensional spaces; and (3) non-negative matrix factorization based methods [13, 14, 21], aiming to minimize reconstruction error between existing data and factorized matrices.

Graph learning-based IMVC represent data using graph structures to mine relationships between views and learn low-dimensional representations from diverse graphs, elucidating relationships among multiple views [38]. Based on the approach used to integrate graph information, graph learning-based IMVC can be categorized into two categories: (1) spectral learning-based methods [37, 40]. These techniques fuse nearest neighbor graph structure and graph regularization into incomplete graph learning, enhancing information exploration. For example, Wen et al. [37] proposed confidence graph learning, inferring missing edges from shared similar-nearest neighbors. (2) adaptive graph learning-based methods [12, 16, 18, 22, 36, 49]. These methods enhance clustering by optimizing graph structures and integrating multi-view information, addressing missing data and learning low-dimensional representations [43]. Through weighting and anchor points strategy, these methods considers the contribution of each view to clustering and reduces the size of data and complexity. For example, Wang et al. [36] constructed individual incomplete bipartite graphs for each view, and treating incomplete samples as unconnected to anchors within the graph. However, despite the advancements in graph-based methods, significant challenges persist in large-scale IMVC. Firstly, the inherent limitation of the first-order affinity graph accuracy leads to misclassification of intra-cluster and inter-cluster samples. Moreover, data incompleteness exacerbates the degradation of the graph, making accurate imputation challenging. Secondly, the computational cost becomes prohibitive, particularly during operations like eigendecomposition or matrix inversion on large datasets.

Deep learning-based IMVC utilizes a deep learning model to learn feature representations and is better able to handle missing data. For example, Xu et al. [44] acquired view-specific features through individual auto-encoders and employed a feature projection technique to explore the consensus information among multiple views.

## 3 The Proposed Method

### 3.1 Notations

Given the incomplete data matrices $\{X^r\}_{r=1}^{v} \in \mathbb{R}^{n \times d^r}$, where $v$ and $n$ is the number of all views and all samples, and $d^r$ is the number of feature dimension of the $r$-th view. Since the incomplete multi-view data [46] under discussion is the missing of random samples in each view of multiple views, and each sample is guaranteed to be observable in at least one view, we can divide the data matrix for each view into observed and missing parts, i.e., $X^r = \{X_o^r, X_m^r\}$, where $X_o^r \in \mathbb{R}^{n^r \times d^r}$ and $X_m^r \in \mathbb{R}^{(n-n^r) \times d^r}$, and $n^r$ denote the number of observed samples of the $r$-th view.

### 3.2 View Independent Bipartite Graph Construction

To leverage the benefits of bipartite graph-based methods for large scale IMVC, we incorporate bipartite graphs to capture the first-order sample-anchor affinities within the FSIMVC-OF model. This

generally involves two subsequent steps: anchor selection and graph construction.

Considering that constructing multiple anchor graphs through diversification mechanisms can enhance the performance of subsequent clustering [48], we adopt a view-independent anchor selection strategy in this paper. Specifically, we employ the cluster centers obtained from k-means applied to complete samples $\mathbf{X}_o^r$ in each incomplete view to form the anchor matrix $\mathbf{A}^r \in \mathbb{R}^{m^r \times d^r}$, where $m^r$ represents the number of anchors in the $r$-th view. This strategy aims to handle the incompleteness in data across different views while enabling independent anchor selection for each view.

With the obtained anchor matrix $\mathbf{A}^r$, the bipartite graph $\mathbf{B}^r \in \mathbb{R}^{n^r \times m^r}$ between observed samples and anchors can be learned by solving the following optimization problem,

$$\min_{b_i^r \mathbf{1}=1, b_i^r \geq 0} \quad \sum_{j=1}^{m^r} h\left(\mathbf{x}_i^r, \mathbf{a}_j^r\right) b_{ij}^r + \gamma \sum_{j=1}^{m^r} \left(b_{ij}^r\right)^2, \tag{1}$$

where $b_i^r$ represents the $i$-th row of $\mathbf{B}^r$, and $h\left(\mathbf{x}_i^r, \mathbf{a}_j^r\right)$ signifies the Euclidean distance between the $i$-th sample and the $j$-th anchor, $\gamma = \frac{k}{2} h(\mathbf{x}_i^r, \mathbf{a}_{(k+1)}^r) - \frac{1}{2} \sum_{j=1}^{k} h(\mathbf{x}_i^r, \mathbf{a}_j^r)$ is the trade off parameter. To ensure sparsity in the bipartite graph $\mathbf{B}^r$ and avoid the search of $\gamma$, we limit each row to retain only $k$ non-zero elements, which is set to $k = 5$ in this paper. The closed-form solution is given by [29]:

$$b_{ij}^r = \begin{cases} \frac{h(\mathbf{x}_i^r, \mathbf{a}_{(k+1)}^r) - h(\mathbf{x}_i^r, \mathbf{a}_j^r)}{kh(\mathbf{x}_i^r, \mathbf{a}_{(k+1)}^r) - \sum_{k'=1}^{k} h(\mathbf{x}_i^r, \mathbf{a}_{k'}^r)}, & j \leq k, \\ 0, & j > k. \end{cases} \tag{2}$$

It is widely acknowledged that such affinity graphs may still incorrectly assign a zero affinity ($b_{ij}^r = 0$) to out-of-neighborhood samples within the same cluster, while neighboring inter-cluster samples may be mistakenly treated as positive [26].

## 3.3 Enhancement via Duality Optimal Graph Filtering

To address the inaccuracies of first-order affinities between samples and anchors, we utilize sample-anchor-sample similarities to capture higher-order interactions among sample-sample. Specifically, given the acquired sample-anchor bipartite graph $\mathbf{B}^r \in \mathbb{R}^{n^r \times m^r}$, we first introduce a diagonal matrix $\triangle^r \in \mathbb{R}^{m^r \times m^r}$, where its $j$-th diagonal element $\triangle_{jj}^r = \sum_{i=1}^{n^r} b_{ij}^r$. Next, we obtain a column-normalized affinity graph $\mathbf{P}^r = \mathbf{B}^r (\triangle^r)^{-\frac{1}{2}}$, and then the affinity graph between sample-anchor-sample can be derived as $\mathbf{S}^r = \mathbf{B}^r \triangle^{-1} \mathbf{B}^{rT} = \mathbf{P}^r \mathbf{P}^{rT}$, where $\mathbf{S}^r \in \mathbb{R}^{n^r \times n^r}$. It can be verified that $\mathbf{S}^r$ is a doubly stochastic matrix, i.e., $\left(\mathbf{P}^r \mathbf{P}^{rT}\right) \mathbf{1}_{n^r} = \mathbf{1}_{n^r}, \mathbf{1}_{n^r}^T \left(\mathbf{P}^r \mathbf{P}^{rT}\right) = \mathbf{1}_{n^r}^T$.

Smoother graph signals, as indicated by [4, 9, 27, 42, 52], correlate with a clearer clustering structure. To achieve a smoother graph [32], we solve the optimization problem:

$$\min_{\bar{\mathbf{P}}^r} \quad ||\bar{\mathbf{P}}^r - \mathbf{P}^r||^2 + \lambda \mathrm{tr}(\bar{\mathbf{P}}^{rT} \mathbf{L}_n^r \bar{\mathbf{P}}^r), \tag{3}$$

where $\mathbf{L}_n^r = \mathbf{I}_{n^r} - \mathbf{P}^r \mathbf{P}^{rT} \in \mathbb{R}^{n^r \times n^r}$ is the normalized Laplacian matrix, and the solution is given by:

$$\bar{\mathbf{P}}^r = (\mathbf{I}_{n^r} + \lambda \mathbf{L}_n^r)^{-1} \mathbf{P}^r. \tag{4}$$

Compared to $\mathbf{P}^r$, the induced graph $\bar{\mathbf{P}}^r$ obtained with Eq. (4) become more smooth by incorporating sample-sample similarities [5, 6, 50, 51]. However, the operator in Eq.(4) requires an inverse operation. This approach is suboptimal for the downstream IMVC task and does not leverage complementary information from other views.

To avoid the inverse operation, the above solution can be approximated through its first-order Taylor expansion, and we have,

$$\bar{\mathbf{P}}^r = (\mathbf{I}_{n^r} - \lambda \mathbf{L}_n^r) \mathbf{P}^r. \tag{5}$$

Recent studies in [30] emphasize the importance of low-frequency bases in smooth signals. To incorporate this insight, we build the following filtered signal

$$\bar{\mathbf{P}}^r = \left(\frac{\mathbf{I}_{n^r} + \mathbf{P}^r \mathbf{P}^{rT}}{2}\right)^t \mathbf{P}^r, \tag{6}$$

To circumvent the need for selecting different values of $t$, we design a learnable graph filter. This filter dynamically updates during the clustering process:

$$\bar{\mathbf{P}}^r = \sum_{t=0}^{\bar{t}} \beta_t^r \left(\frac{\mathbf{I}_{n^r} + \mathbf{P}^r \mathbf{P}^{rT}}{2}\right)^t \mathbf{P}^r = \mathcal{H}(\mathbf{P}^r, \boldsymbol{\beta}^r)\mathbf{P}^r. \tag{7}$$

where $t$ represents a positive integer that defines the extent of the $t$-hop neighborhood relationship within the signal. From Eq. (7), it can be seen that the first order bipartite graph $\mathbf{P}^r$ is smoothed by a high-order graph filter $\mathcal{H}(\mathbf{P}^r, \boldsymbol{\beta}^r) \in \mathbb{R}^{n^r \times n^r}$ with the unknown coefficient $\boldsymbol{\beta}^r$. Compared to Eq. (4), it can be conclude that Eq. (7) leverage optimal graph filter learning on the sample side to encapsulate higher-order interactions, guiding the enhancement of clustering-friendly structures for the first-order bipartite graph. However, the computation of Eq. (7) is still intensive.

It can be further proven by Theorem 3 that the filtering on $\mathbf{P}^r$ by sample-side graph filter can be equivalently represented by the filtering on $\mathbf{P}^{rT}$ by the anchor-side graph filter. With such equivalence, we propose the following filtering to incorporate the clustering friendly high order sample-sample interactions via the high order anchor-anchor interactions,

$$\bar{\mathbf{P}}^r = \mathbf{P}^r \sum_{t=0}^{\bar{t}} \beta_t^r \left(\frac{\mathbf{I}_{m^r} + \mathbf{P}^{rT} \mathbf{P}^r}{2}\right)^t = \mathbf{P}^r \mathcal{H}(\mathbf{P}^{rT}, \boldsymbol{\beta}^r). \tag{8}$$

Compared with the sample-side filtering in Eq. (7), the anchor-side filtering in Eq. (8) maintains adaptive filter learning on the anchor side, ensuring computational efficiency with linear complexity on the sample size, based on the provable equivalence between these two types of graph filters.

## 3.4 Consensus Clustering with Smooth Bipartite Graphs

Given multiple incomplete first-order bipartite graphs $\{\mathbf{P}^r\}_{r=1}^v$ with $\mathbf{P}^r \in \mathbb{R}^{n^r \times m^r}$, we take the aforementioned anchor-side optimal graph filter to incorporate the higher order interactions among samples. In this subsection, we aim to learn a complete and consensus clustering oriented representation $\mathbf{Z} \in \mathbb{R}^{n \times c}$ from multiple incomplete bipartite graphs, each enhanced by view-independent graph filters, denoted as $\{\mathbf{P}^r \mathcal{H}(\mathbf{P}^{rT}, \boldsymbol{\beta}^r)\}_{r=1}^v$. Based on the indices of observed and missing samples in each view of the incomplete

data matrices $\{\mathbf{X}^r\}_{r=1}^v$, we can partition the consensus sample matrix $\mathbf{Z}$ into two corresponding parts for each view, represented as $\mathbf{Z} = \{\mathbf{Z}_{o^r}, \mathbf{Z}_{m^r}\}$. We propose to project anchor points from different views into a common cluster latent space using projection matrices $\{\mathbf{W}^r\}_{r=1}^v \in \mathbb{R}^{m^r \times c}$, where $c$ is the cluster number. These points are linked with $c$ prototypes in the latent space using the prototype representation matrix $\mathbf{C} \in \mathbb{R}^{c \times c}$. The consensus clustering procedure with the enhanced bipartite graphs of FSIMVC-OF can be formulated as follows:

$$\min \quad \sum_{r=1}^v (\alpha^r)^2 \left\| \mathbf{P}^r \left( \sum_{t=0}^{\bar{t}} \beta_t^r \left( \frac{\mathbf{I}_{m^r} + \mathbf{P}^{rT}\mathbf{P}^r}{2} \right)^t \right) - \mathbf{Z}_{o^r} \mathbf{C}\mathbf{W}^{rT} \right\|_F^2 \quad (9)$$

$$\text{s.t.} \quad \mathbf{Z} \geq 0, \mathbf{1}_n^T \mathbf{Z} = \mathbf{1}_c^T, \mathbf{W}^{rT}\mathbf{W}^r = \mathbf{I}_c, \mathbf{C}^T\mathbf{C} = \mathbf{I}_c,$$
$$\boldsymbol{\alpha}^T \mathbf{1} = 1, 0 \leq \alpha^r \leq 1, \boldsymbol{\beta}^{rT}\mathbf{1} = 1, 0 \leq \beta_t^r \leq 1,$$

where $\alpha^r$ is the view weight for the $r$-th view, $\mathbf{Z}_{o^r}$ represents the similarity between $c$ prototypes and $n^r$ observed samples from the $r$-th view. To enhance the distinctiveness of the learned prototypes $\mathbf{C}$, we apply orthogonal constraints to $\mathbf{C}$. The prototype graph $\mathbf{Z}$, thus learned, must fulfill $\mathbf{Z} \geq 0$ and $\mathbf{1}_n^T \mathbf{Z} = \mathbf{1}_c^T$ conditions.

## 3.5 Optimization

The problem in Eq.(9) involves five variables: $\mathbf{W}^r, \mathbf{C}, \mathbf{Z}, \boldsymbol{\beta}^r$ and $\boldsymbol{\alpha}$. We provide an alternative algorithm for optimization.

*3.5.1 Update $\mathbf{W}^r$.* When other variables are fixed, the problem for $\mathbf{W}^r$ becomes:

$$\min_{\mathbf{W}^r} \quad \left\| \mathbf{P}^r \left( \sum_{t=0}^{\bar{t}} \beta_t^r \mathbf{Q}_t^r \right) - \mathbf{Z}_{o^r} \mathbf{C}\mathbf{W}^{rT} \right\|_F^2, \quad \text{s.t.} \quad \mathbf{W}^{rT}\mathbf{W}^r = \mathbf{I}_c, \quad (10)$$

where $\mathbf{Q}_t^r = \left( \frac{\mathbf{I}_{m^r} + \mathbf{P}^{rT}\mathbf{P}^r}{2} \right)^t$, which can be further simplified as

$$\max_{\mathbf{W}^r} \quad \text{tr}(\mathbf{W}^{rT}\mathbf{E}^r), \quad \text{s.t.} \quad \mathbf{W}^{rT}\mathbf{W}^r = \mathbf{I}_c, \quad (11)$$

where $\mathbf{E}^r = \mathbf{F}^{rT}\mathbf{P}^{rT}\mathbf{Z}_{o^r}\mathbf{C}$, and $\mathbf{F}^r = \sum_{t=0}^{\bar{t}} \beta_t^r \mathbf{Q}_t^r$. The optimal $\mathbf{W}^r$ can be obtained by Singular Value Decomposition (SVD) on $\mathbf{E}^r$ [34].

*3.5.2 Update $\mathbf{C}$.* Similar to Eq. (11), we seek to optimize $\mathbf{C}$ via:

$$\max_{\mathbf{C}} \quad \text{tr}(\mathbf{C}^T\mathbf{R}), \quad \text{s.t.} \quad \mathbf{C}^T\mathbf{C} = \mathbf{I}_c, \quad (12)$$

where $\mathbf{R} = \sum_{r=1}^v (\alpha^r)^2 \mathbf{Z}_{o^r}^T \mathbf{P}^r \mathbf{F}^r \mathbf{W}^r$. Similar to the update of $\mathbf{W}^r$, the optimization of $\mathbf{C}$ can also be performed by the SVD on $\mathbf{R}$.

*3.5.3 Update $\mathbf{Z}$.* To address the sub-problem of $\mathbf{Z}$, we begin by introducing

$$\mathbf{H}_{o^r} = \left( (\alpha^r)^2 \mathbf{P}^r \mathbf{F}^r \mathbf{W}^r \mathbf{C} \right) / \sum_{r=1}^v (\alpha^r)^2, \quad (13)$$

where $\mathbf{H}_{o^r} \in \mathbb{R}^{n^r \times c}$, and $\mathbf{H} \in \mathbb{R}^{n \times c}$ can be obtained by aggregating all $\{\mathbf{H}_{o^r}\}_{r=1}^v$ at the corresponding sample index. Consequently, the optimization problem related to $\mathbf{Z}$ can be decomposed row-wise. For the $i$-th sample, it is formulated as:

$$\min_{\mathbf{z}_i} \quad \|\mathbf{z}_i - \mathbf{h}_i\|_F^2 \quad \text{s.t.} \quad \mathbf{1}_n^T \mathbf{z}_i = 1, \quad \mathbf{z}_i \geq 0. \quad (14)$$

The above problem in Eq. (14) can be efficiently solved using the Euclidean projection onto the simplex [28].

*3.5.4 Update $\boldsymbol{\beta}^r$.* The rest problem w.r.t. $\boldsymbol{\beta}^r \in \mathbb{R}^{\bar{t} \times 1}$ can be written as:

$$\min_{\boldsymbol{\beta}^r} \quad \boldsymbol{\beta}^{rT}\mathbf{M}^r\boldsymbol{\beta}^r - 2\boldsymbol{\beta}^{rT}\mathbf{s}^r \quad \text{s.t.} \quad \boldsymbol{\beta}^{rT}\mathbf{1} = 1, 0 \leq \beta_t^r \leq 1, \quad (15)$$

where $\mathbf{M}^r \in \mathbb{R}^{\bar{t} \times \bar{t}}$ with $\mathbf{M}_{ij}^r = \text{tr}(\mathbf{Q}_i^r \mathbf{P}^{rT}\mathbf{P}^r \mathbf{Q}_j^r)$, and $\mathbf{s}^r \in \mathbb{R}^{\bar{t} \times 1}$ with $s_t^r = \text{tr}(\mathbf{Q}_t^r \mathbf{P}^{rT}\mathbf{Z}_{o^r}\mathbf{C}\mathbf{W}^{rT})$. Eq. (15) can be readily solved by off-the-shelf quadratic programming solvers.

*3.5.5 Update $\boldsymbol{\alpha}$.* The optimization problem for $\boldsymbol{\alpha}$ can be formulated as follows:

$$\min_{\boldsymbol{\alpha}} \quad \sum_{r=1}^v (\alpha^r)^2 u^r, \quad \text{s.t.} \quad \boldsymbol{\alpha}^T\mathbf{1} = 1, 0 \leq \alpha^r \leq 1, \quad (16)$$

where $u^r = \left\| \mathbf{P}^r (\sum_{t=0}^{\bar{t}} \beta_t^r \mathbf{Q}_t^r) - \mathbf{Z}_{o^r}\mathbf{C}\mathbf{W}^{rT} \right\|_F^2$. The values for $\boldsymbol{\alpha}$ can be determined using the Cauchy-Schwarz inequality:

$$\alpha^r = \frac{1/u^r}{\sum_{r'=1}^v (1/u^{r'})}. \quad (17)$$

The procedure of FSIMVC-OF is encapsulated within Algorithm 1.

---

**Algorithm 1** Algorithm for FSIMVC-OF.

---

**Input:** Incomplete dataset $\{\mathbf{X}^r\}_{r=1}^v \in \mathbb{R}^{n^r \times d^r}$, the anchor numbers of each views $\{m^r\}_{r=1}^v$, and the cluster number $c$.
1: Generating anchors $\{\mathbf{A}^r\}_{r=1}^v$ for all views by k-means;
2: Constructing bipartite graphs $\{\mathbf{B}^r\}_{r=1}^v$ for all views by Eq. (2);
3: Calculate the high-order graphs $\{\mathbf{Q}_t^r\}_{r=1,t=0}^{v,\bar{t}}$ for all views;
4: **Initialization:** $\{\mathbf{W}^r\}_{r=1}^v, \mathbf{C}, \mathbf{Z}, \boldsymbol{\alpha}, \{\boldsymbol{\beta}^r\}_{r=1}^v$;
5: **repeat**
6:     Update $\{\mathbf{W}^r\}_{r=1}^v$ by solving Eq. (11);
7:     Update $\mathbf{C}$ by solving Eq. (12);
8:     Update $\mathbf{Z}$ by solving Eq. (14);
9:     Update $\{\boldsymbol{\beta}^r\}_{r=1}^v$ by solving Eq. (15);
10:    Update $\boldsymbol{\alpha}$ by Eq. (17);
11: **until** Converges
**Output:** Obtain clustering result from $\mathbf{Z}$.

---

## 3.6 Convergence and Complexity

The objective function of FSIMVC-OF has a lower bound of zero. Through decomposing Eq. (9) into convex sub-problems, each with a globally optimal solution, the alternating optimization strategies ensure a monotonic decrease in its objective function value until convergence, in accordance with principles outlined in [2].

The computational complexity of FSIMVC-OF encompasses four aspects, as previously mentioned. Generating anchors and constructing bipartite graphs have complexities of $O(t_1 \sum_{r=1}^v n^r m^r d^r)$ and $O(k \sum_{r=1}^v n^r m^r)$ respectively. Computing the high-order graph filter requires $O(\sum_{r=1}^v (n^r (m^r)^2) + \bar{t}(m^r)^3)$ time. Updating all variables entails a complexity of $O(t_2 c^2 \sum_{r=1}^v n^r m^r)$. Here, $t_1$ and $t_2$ denote the numbers of iterations for anchor generation and variable updates. Considering $t_1, t_2, \bar{t}, m^r, c \ll n^r < n$, the overall computational complexity of FSIMVC-OF remains $O(n)$. Thereby, our method efficiently attains IMVC through linear complexity.

# 4 Theoretical Analysis

In this section, we delve into the theoretical underpinnings of how the sample-side optimal graph filter $\mathcal{H}(\mathbf{P}^r, \boldsymbol{\beta}^r)$ and anchor-side optimal graph filter $\mathcal{H}(\mathbf{P}^{rT}, \boldsymbol{\beta}^r)$ contribute to enhancing clustering performance and prove the equivalence between them.

As elucidated in [10], the eigenvalues of a graph Laplacian matrix are indicative of its structural properties: low eigenvalues correspond to overarching features like clusters, while high eigenvalues capture finer details and noise. Thus, for optimal clustering, it is essential to employ a low-pass graph filter capable of attenuating high eigenvalues while preserving the low ones. Consider a graph with a Laplacian matrix $\mathbf{L}$ possessing $n$ eigenvalues $\lambda_1 \leq \lambda_2 \leq \ldots \leq \lambda_n$. Let $\mathcal{H}(\cdot) : \mathbb{R}^{n \times n} \to \mathbb{R}^{n \times n}$ denote a transformation applied to the Laplacian matrix, resulting in eigenvalues $h(\lambda_1), \ldots, h(\lambda_n)$, where $h(\cdot)$ represents the transformation function. Wai et al. provide the following definition of a low-pass graph filter [35]:

**Definition 1.** [35] (Low-pass graph filter) $\mathcal{H}(\mathbf{L})$ is deemed a $(K, \eta)$ low-pass graph filter if:

$$\eta := \frac{\max(|h(\lambda_{K+1})|, |h(\lambda_{K+2})|, \cdots, |h(\lambda_n)|)}{\min(|h(\lambda_1)|, |h(\lambda_2)|, \cdots, |h(\lambda_K)|)} \in [0, 1), \quad (18)$$

Here, $\eta$ acts as the low-pass coefficient, indicating that for a graph filter $\mathcal{H}(\mathbf{L})$, if there exists an integer $1 \leq K < n$ and a coefficient $\eta < 1$, $\mathcal{H}(\mathbf{L})$ is considered a low-pass graph filter.

Now, we demonstrate that the sample-side graph filter $\mathcal{H}(\mathbf{P}^r, \boldsymbol{\beta}^r)$ in Eq. (7) is a low-pass graph filter, as proven in Theorem 1.

**Theorem 1.** The sample-side optimal graph filter $\mathcal{H}(\mathbf{P}^r, \boldsymbol{\beta}^r) = \sum_{t=0}^{\bar{t}} \beta_t^r \left( \frac{\mathbf{I}_{n^r} + \mathbf{P}^r \mathbf{P}^{rT}}{2} \right)^t$ with $\boldsymbol{\beta}^{rT} \mathbf{1} = 1$ and $0 \leq \beta_t^r \leq 1$ is a $(K, \eta)$ low-pass graph filter for $1 < K \leq m^r$.

PROOF. Let $\mathbf{P}^r \in \mathbb{R}^{n^r \times m^r}$ be the bipartite graph with $m^r < n^r$. The singular values of $\mathbf{P}^r$ are $1 \geq \sigma_1 \geq \sigma_2 \geq \ldots \geq \sigma_{m^r}$, and the top-$m^r$ largest eigenvalues of $\left( \mathbf{P}^r \mathbf{P}^{rT} \right)$ are $1 \geq \sigma_1^2 \geq \sigma_2^2 \geq \ldots \geq \sigma_{m^r}^2$. Thus, the eigenvalues of $\mathbf{L}^r = \mathbf{I}_{n^r} - \mathbf{P}^r \mathbf{P}^{rT}$ are $0 \leq (1 - \sigma_1^2) \leq (1 - \sigma_2^2) \leq \ldots \leq (1 - \sigma_{m^r}^2) \leq 1 = \ldots = 1$, denoted as $\{\lambda_1, \lambda_2, \ldots, \lambda_{n^r}\}$, satisfying $0 = \lambda_1 \leq \lambda_2 \leq \ldots \leq \lambda_{n^r} \leq 1$. Then, the singular values of $\mathbf{P}^r$ are $\left\{ \sqrt{1 - \lambda_1}, \sqrt{1 - \lambda_2}, \ldots, \sqrt{1 - \lambda_{m^r}} \right\}$. Similarly, the eigenvalues of $\mathcal{H}(\mathbf{P}^r, \boldsymbol{\beta}^r) = \sum_{t=0}^{\bar{t}} \beta_t^r \left( \frac{\mathbf{I}_{n^r} + \mathbf{P}^r \mathbf{P}^{rT}}{2} \right)^t$ are $\left\{ \sum_{t=0}^{\bar{t}} \beta_t^r (\frac{2-\lambda_1}{2})^t, \sum_{t=0}^{\bar{t}} \beta_t^r (\frac{2-\lambda_2}{2})^t, \ldots, \sum_{t=0}^{\bar{t}} \beta_t^r (\frac{2-\lambda_{n^r}}{2})^t \right\}$. Hence, we compute its low-pass coefficient $\eta$ as defined in Definition 1:

$$\eta = \frac{\max \left( \left| \sum_{t=0}^{\bar{t}} \beta_t^r (\frac{2-\lambda_{K+1}}{2})^t \right|, \cdots, \left| \sum_{t=0}^{\bar{t}} \beta_t^r (\frac{2-\lambda_{n^r}}{2})^t \right| \right)}{\min \left( \left| \sum_{t=0}^{\bar{t}} \beta_t^r (\frac{2-\lambda_1}{2})^t \right|, \cdots, \left| \sum_{t=0}^{\bar{t}} \beta_t^r (\frac{2-\lambda_K}{2})^t \right| \right)} \quad (19)$$

$$= \frac{\sum_{t=0}^{\bar{t}} \beta_t^r (\frac{2-\lambda_{K+1}}{2})^t}{\sum_{t=0}^{\bar{t}} \beta_t^r (\frac{2-\lambda_K}{2})^t}.$$

Given the non-zero singular values of $\mathbf{P}^r$, denoted by $\sigma_K \geq 0$, the corresponding eigenvalues of the matrix $\mathbf{L}^r$ are ordered that $\lambda_K < \lambda_{K+1}$. This ordering implies that for any positive integer $t$, the inequality $0 < (\frac{2-\lambda_{K+1}}{2})^t < (\frac{2-\lambda_K}{2})^t$ holds. By analyzing the vector $\boldsymbol{\beta}^r$ with $\boldsymbol{\beta}^{rT} \mathbf{1} = 1$ and $0 \leq \beta_t^r \leq 1$, we can observe that: $\sum_{t=0}^{\bar{t}} \beta_t^r \left( \frac{2-\lambda_{K+1}}{2} \right)^t < \sum_{t=0}^{\bar{t}} \beta_t^r \left( \frac{2-\lambda_K}{2} \right)^t$, which implies $\eta < 1$.

According to Definition 1, the learned graph filter $\mathcal{H}(\mathbf{P}^r, \boldsymbol{\beta}^r) = \sum_{t=0}^{\bar{t}} \beta_t^r \left( \frac{\mathbf{I}_{n^r} + \mathbf{P}^r \mathbf{P}^{rT}}{2} \right)^t$ can be classified as a low-pass graph filter. □

Next, we further show that the anchor-side optimal graph filter $\mathcal{H}(\mathbf{P}^{rT}, \boldsymbol{\beta}^r)$ in Eq. (8) is also a low-pass graph filter by Theorem 2.

**Theorem 2.** The anchor-side optimal graph filter $\mathcal{H}(\mathbf{P}^{rT}, \boldsymbol{\beta}^r) = \sum_{t=0}^{\bar{t}} \beta_t^r \left( \frac{\mathbf{I}_{m^r} + \mathbf{P}^{rT} \mathbf{P}^r}{2} \right)^t$ with $\boldsymbol{\beta}^{rT} \mathbf{1} = 1$ and $0 \leq \beta_t^r \leq 1$ is also a low-pass graph filter.

PROOF. Similar to the sample-side graph filter learning, utilizing the anchor-side Laplacian matrix $\bar{\mathbf{L}} = \mathbf{I}_{m^r} - \mathbf{P}^{rT} \mathbf{P}^r$, we establish that the anchor-side graph filter $\mathcal{H}(\mathbf{P}^{rT}, \boldsymbol{\beta}^r) = \sum_{t=0}^{\bar{t}} \beta_t^r \left( \frac{\mathbf{I}_{m^r} + \mathbf{P}^{rT} \mathbf{P}^r}{2} \right)^t$ with $\boldsymbol{\beta}^{rT} \mathbf{1} = 1$ and $0 \leq \beta_t^r \leq 1$ is also a low-pass graph filter. Detailed proof is omitted due to space constraints. □

Subsequently, we establish the equivalence between the sample-side graph filter $\mathcal{H}(\mathbf{P}^{rT}, \boldsymbol{\beta}^r)$ applied to the sample-side of bipartite graph $\mathbf{P}^r$ and the anchor-side graph filter $\mathcal{H}(\mathbf{P}^{rT}, \boldsymbol{\beta}^r)$ operated on the anchor-side of $\mathbf{P}^r$ in Theorem 3.

**Theorem 3.** The process of filtering the graph through sample-side graph filtering, denoted by $\mathcal{H}(\mathbf{P}^r, \boldsymbol{\beta}^r) \mathbf{P}^r$, yields an equivalent result to that of filtering by anchor-side graph filtering, represented as $\mathbf{P}^r \mathcal{H}(\mathbf{P}^{rT}, \boldsymbol{\beta}^r)$. This equivalence is formally expressed as:

$$\mathcal{H}(\mathbf{P}^r, \boldsymbol{\beta}^r) \mathbf{P}^r = \mathbf{P}^r \mathcal{H}(\mathbf{P}^{rT}, \boldsymbol{\beta}^r). \quad (20)$$

PROOF. Let the SVD decomposition of $\mathbf{P}^r$ be $\mathbf{P}^r = \mathbf{U}^r \Sigma^r \mathbf{V}^{rT}$, where $\mathbf{U}^r \in \mathbb{R}^{n^r \times n^r}$, $\Sigma^r \in \mathbb{R}^{n^r \times m^r}$, and $\mathbf{V}^r \in \mathbb{R}^{m^r \times m^r}$. In terms of the Graph Fourier Transform (GFT), the sample-side graph filter $\mathcal{H}(\mathbf{P}^r, \boldsymbol{\beta}^r) = \sum_{t=0}^{\bar{t}} \beta_t^r \left( \frac{\mathbf{I}_{n^r} + \mathbf{P}^r \mathbf{P}^{rT}}{2} \right)^t$ transforms a sample-side graph signal on $n^r$ vertices using the projection matrix $\mathbf{U}^r$, with the corresponding frequency response function given by $h(\lambda_K) = \sum_{t=0}^{\bar{t}} \beta_t^r \left( \frac{2-\lambda_K}{2} \right)^t$. Notably, $h(\lambda_K) = \sum_{t=0}^{\bar{t}} \beta_t^r (\frac{1+\sigma_K^2}{2})^t = h(\sigma_K)$ for the top-$m^r$ singular values. Similarly, the anchor-side graph filter $\mathcal{H}(\mathbf{P}^{rT}, \boldsymbol{\beta}^r) = \sum_{t=0}^{\bar{t}} \beta_t^r \left( \frac{\mathbf{I}_{m^r} + \mathbf{P}^{rT} \mathbf{P}^r}{2} \right)^t$ transforms an anchor-side graph signal on $m^r$ vertices using the projection matrix $\mathbf{V}^r$, with the frequency response function given by $h(\sigma_K) = \sum_{t=0}^{\bar{t}} \beta_t^r \left( \frac{1+\sigma_K^2}{2} \right)^t$. From the above analysis, the following equation holds:

$$\mathcal{H}(\mathbf{P}^r, \boldsymbol{\beta}^r) \mathbf{P}^r \quad (21)$$

$$= \mathbf{U}^r \begin{bmatrix} h(\sigma_1) & & & & & \\ & \ddots & & & & \\ & & h(\sigma_{m^r}) & & & \\ & & & h(\lambda_{m^r+1}) & & \\ & & & & \ddots & \\ & & & & & h(\lambda_{n^r}) \end{bmatrix} \mathbf{U}^{rT} \mathbf{U}^r \begin{bmatrix} \sigma_1 & & \\ & \ddots & \\ 0 & \ldots & \sigma_{m^r} \\ & & 0 \\ \vdots & & \vdots \\ 0 & \ldots & 0 \end{bmatrix} \mathbf{V}^{rT}$$

$$= \mathbf{U}^r \begin{bmatrix} h(\sigma_1)\sigma_1 & & \\ & \ddots & \\ & & h(\sigma_{m^r})\sigma_{m^r} \\ 0 & \ldots & 0 \\ \vdots & & \vdots \\ 0 & \ldots & 0 \end{bmatrix} \mathbf{V}^{rT} = \mathbf{U}^r \Sigma^r \mathbf{V}^{rT} \mathbf{V}^r \begin{bmatrix} h(\sigma_1) & & \\ & \ddots & \\ & & h(\sigma_{m^r}) \end{bmatrix} \mathbf{V}^{rT}$$

$$= \mathbf{P}^r \mathcal{H}(\mathbf{P}^{rT}, \boldsymbol{\beta}^r).$$

□

This equation emphasizes the symmetry in filtering, suggesting that filtering through either the sample side or the anchor side produces the same result.

## 5 Experiments

### 5.1 Benchmark Datasets

Nine prevalent multi-view benchmark datasets were utilized in our experiments, comprising COIL20, Handwritten [1], BDGP, Scene-15 [8], MNIST-10K [15], ALOI-100 [11], Reuters, YTF-10 and FMNIST. YTF-10 is a subset of face videos obtained from YouTube. Table 1 provides a summary of these datasets.

**Table 1: Summary of the datasets.**

| ID | Dataset | View | Size | Class | Feature |
|----|---------|------|------|-------|---------|
| D1 | COIL20 | 4 | 1440 | 20 | 1024/944/4096/576 |
| D2 | Handwritten | 6 | 2000 | 10 | 240/76/216/47/64/6 |
| D3 | BDGP | 3 | 2500 | 5 | 1000/500/250 |
| D4 | Scene-15 | 3 | 4485 | 15 | 20/59/40 |
| D5 | MNIST-10K | 3 | 10000 | 10 | 30/9/30 |
| D6 | ALOI-100 | 4 | 10800 | 100 | 77/13/64/125 |
| D7 | Reuters | 5 | 18758 | 6 | 21531/24892/34251/15506/11547 |
| D8 | YTF-10 | 4 | 38654 | 10 | 944/576/512/640 |
| D9 | FMNIST | 3 | 60000 | 10 | 512/512/1280 |

### 5.2 Compared Methods and Settings

To demonstrate the superiority of FSIMVC-OF, we compare it with ten state-of-the-art IMVC algorithms including DAIMC [14], SRLC [53], LF-IMVC [24], EE-R-IMVC [23], IMVC-CBG [36], SAGF-IMC [18], SGC-IMVC [20], LI-MKKM-MR [17], PSIMVC-PG [16] and FIMVC-VIA [22]. These methods include NMF-based methods, bipartite graph-based methods, kernel-based methods, graph filter-based methods and so on [39].

For each dataset, we generate incomplete versions with missing ratio $\varepsilon = \frac{n-n_p}{n}$ varying as $[0.1 : 0.2 : 0.9]$ according to IMVC-CBG [36]. To ensure consistent and fair evaluation, we employ the original code provided by the authors and rigorously follow the suggested settings and parameter search methods for all baseline approaches. Specifically, for each dataset, we generate 10 unique sets of incomplete data corresponding to each defined missing ratio. We then apply each clustering method to these predefined sets. The effectiveness of these methods is assessed by computing the average clustering performance across these 10 sets.

To evaluate the clustering performances of different IMVC methods, we employ four well-established evaluation metrics: accuracy (ACC) [41], Normalized Mutual Information (NMI) [33] and Purity [31]. For all the aforementioned evaluation metrics, higher values indicate better clustering performance.

For FSIMVC-OF, the $k$-nearest neighbor parameter $k$ is fixed to 5 in the entire experiments. The order of graph filter $\bar{t}$ is fixed to 6, and for comparison purposes, the number of anchors for different views are set as $m^1 = m^2 = ... = m^v = m$, where $m$ is searched in $\{2c, 4c, 6c, 8c\}$.

### 5.3 Experimental Results

The comparative analysis in Table 2 reveals the average clustering performance across different missing data ratios. It presents the mean and standard deviation (std) for each method across all datasets and missing ratios, with the top performers marked in red and the runners-up in blue. Methods exceeding a 24-hour computation time are marked with "-", and out-of-memory (OOM) instances are also indicated.

Key observations from the results include: (1) FSIMVC-OF stands out, consistently outperforming other methods in ACC, NMI and Purity metrics for most datasets, with a particularly strong showing on dataset D9, where it significantly outperformed the next best results by 69.92% in ACC, underscoring its strength in large-scale scenarios. (2) Despite being designed for large-scale clustering, bipartite graph-based methods like FIMVC-VIA, PSIMVC-PG, and IMVC-CBG show limited performance, possibly due to their struggle with capturing higher-order data information, an area where FSIMVC-OF excels. (3) While SGC-IMVC demonstrates modest improvements in clustering performance through the application of graph filters on sample-sample graphs, its computational intensity renders it is not suitable for large-scale IMVC tasks. (4) As shown in Figure 1, the superior performance of FSIMVC-OF is consistent across all missing ratios, demonstrating the method's robustness and reliability..

### 5.4 Running Time Comparison

In order to evaluate the computational efficiency of the proposed methods, we recorded the average running times of the baseline algorithms on all benchmark datasets with various missing ratios and report them in Figure 2. The outcomes of certain baseline algorithms on large-scale datasets remain undisclosed due to memory overflow issues. The results suggest that FSIMVC-OF demonstrates the shortest execution running time among all benchmark algorithms across all benchmark datasets, underscoring its superior computational efficiency.

### 5.5 Parameter Sensitivity Analysis

In this section, we delve into the analysis of parameter $m$. Figure 3 showcases the ACC metric of FSIMVC-OF on datasets D2 and D5 with different value of $m$, with a missing ratio of 0.5. Insights gleaned from Fig. 3 reveal that varying values of $m$ demonstrate a limited effect on clustering performance. And with the increase of anchor number, clustering performance shows an upward trend as a whole. Collectively, these findings underscore the reliability and efficacy of FSIMVC-OF across diverse parameter settings.

### 5.6 Convergence Study

Experiments were conducted on datasets D2 and D5 to assess the convergence behavior of our proposed method. The experimental results in terms of the objective function value and ACC of FSIMVC-OF across these datasets are depicted in Fig. 4. We set the algorithm to iterate 30 times, with a fixed missing ratio of 0.1. As illustrated in Fig. 4, the convergence behavior of our algorithm unfolds in a distinct pattern. Initially, the algorithm demonstrates a convergence trend, characterized by a gradual decrease in the objective function value. This is followed by a synchronized pattern

**Table 2: Average clustering performance and comparison (mean±std) of FSIMVC-OF with ten baseline methods on nine datasets.**

| Methods | DAIMC | SRLC | LF-IMVC | EE-R-IMVC | IMVC-CBG | SAGF-IMC | SGC-IMVC | LI-MKKM-MR | PSIMVC-PG | FIMVC-VIA | FSIMVC-OF |
|---|---|---|---|---|---|---|---|---|---|---|---|
| | | | | | ACC(%) | | | | | | |
| D1 | 71.59±2.54 | 74.86±1.09 | 74.58±2.07 | 75.96±1.66 | 64.15±3.48 | 67.54±3.19 | 71.93±7.68 | 68.88±1.71 | 62.10±3.08 | **76.41±2.69** | **80.10±1.64** |
| D2 | 79.82±4.26 | **91.03±1.05** | 85.57±3.28 | 88.03±1.71 | 67.11±3.14 | 83.79±1.44 | 79.50±2.88 | 85.83±0.52 | 29.68±2.49 | 78.71±1.32 | **92.67±1.30** |
| D3 | 37.69±2.56 | 40.42±3.74 | **41.81±2.87** | 40.00±2.85 | 37.83±3.52 | 22.24±2.24 | 25.77±3.58 | OOM | **43.19±2.21** | 39.70±2.44 | 41.46±5.24 |
| D4 | 32.85±1.83 | 33.60±1.48 | 34.81±1.50 | 37.08±1.16 | 26.34±1.00 | 36.68±1.89 | **39.44±1.69** | OOM | 28.51±2.07 | 30.34±1.24 | **39.96±1.62** |
| D5 | 17.65±5.57 | 71.03±3.17 | 66.71±3.36 | 70.74±1.63 | 66.26±2.01 | 72.74±2.35 | - | OOM | 65.59±1.96 | **72.98±2.05** | **79.97±0.43** |
| D6 | 29.58±2.26 | 68.44±0.76 | **70.41±1.03** | 68.33±1.41 | 28.29±0.98 | 56.34±1.24 | - | OOM | 6.44±0.26 | 45.76±1.25 | **71.81±1.52** |
| D7 | OOM | 21.67±1.57 | 37.47±1.36 | 37.16±1.15 | 44.33±1.87 | OOM | - | OOM | 46.75±0.55 | **48.31±0.72** | **50.05±3.67** |
| D8 | 76.19±5.57 | 55.05±3.17 | 78.50±3.36 | **79.28±1.63** | 71.14±2.01 | OOM | - | OOM | 71.81±1.96 | 78.91±2.05 | **81.14±4.24** |
| D9 | OOM | OOM | OOM | OOM | **23.24±1.03** | OOM | OOM | OOM | 22.15±0.58 | 21.26±1.11 | **39.49±4.06** |
| | | | | | NMI(%) | | | | | | |
| D1 | 79.55±1.26 | 81.85±0.91 | 81.10±1.27 | 82.17±1.07 | 73.61±2.38 | 79.08±1.68 | **87.24±3.23** | 75.88±0.90 | 73.64±1.49 | 82.79±1.24 | **86.20±0.87** |
| D2 | 70.46±3.30 | 83.21±1.98 | 76.29±2.23 | 78.49±1.74 | 59.29±2.03 | **85.04±0.97** | 81.57±1.61 | 75.56±0.75 | 26.72±3.21 | 69.04±1.39 | **86.12±1.78** |
| D3 | 13.16±2.93 | 16.59±2.62 | 16.17±2.59 | 16.94±0.77 | 13.99±3.11 | 0.75±0.66 | 2.85±2.87 | OOM | **18.98±1.53** | 15.45±1.88 | **19.18±4.44** |
| D4 | 28.69±1.52 | 29.78±0.96 | 29.02±0.91 | 32.51±0.80 | 21.51±0.82 | 34.09±1.39 | **34.92±0.96** | OOM | 26.18±1.77 | 27.02±1.07 | **36.18±0.86** |
| D5 | 7.81±6.24 | 65.47±1.73 | 54.24±1.04 | 58.49±0.32 | 54.98±1.05 | **69.66±1.04** | - | OOM | 54.52±1.33 | 59.52±0.62 | **67.19±0.59** |
| D6 | 46.43±2.22 | 72.98±0.42 | **76.80±0.35** | 76.77±0.49 | 40.70±1.15 | 66.55±1.10 | - | OOM | 11.93±0.31 | 62.53±0.43 | **78.57±1.04** |
| D7 | OOM | 1.49±0.45 | 16.01±1.21 | 17.77±1.35 | 26.14±3.59 | OOM | - | OOM | 29.44±1.13 | **30.42±0.90** | **34.47±3.49** |
| D8 | 78.01±6.24 | 58.12±1.73 | 79.70±1.04 | 79.44±0.32 | 73.11±1.05 | OOM | - | OOM | 73.35±1.33 | **80.31±0.62** | **83.00±2.82** |
| D9 | OOM | OOM | OOM | OOM | **5.86±0.47** | OOM | OOM | OOM | 3.08±0.19 | 4.63±0.61 | **17.18±2.93** |
| | | | | | Purity(%) | | | | | | |
| D1 | 74.07±2.22 | 77.45±1.19 | 75.78±1.78 | 77.21±1.83 | 66.52±3.29 | 70.79±2.57 | **77.68±6.04** | 71.28±1.32 | 65.75±2.48 | 77.63±2.51 | **83.58±1.42** |
| D2 | 79.93±4.12 | **91.03±1.05** | 85.73±2.77 | 88.03±1.71 | 69.10±2.41 | 85.91±1.15 | 80.95±2.47 | 85.83±0.52 | 30.40±2.55 | 78.71±1.32 | **92.83±1.56** |
| D3 | 38.31±2.52 | 43.35±3.73 | 44.02±3.19 | 42.12±2.69 | 38.73±3.12 | 22.38±2.28 | 26.45±3.91 | OOM | **44.87±1.85** | 40.80±2.05 | **56.79±9.36** |
| D4 | 35.63±1.77 | 37.90±1.46 | 37.25±1.37 | 39.70±1.21 | 27.58±0.89 | 39.84±1.79 | **43.22±1.32** | OOM | 31.36±2.06 | 32.11±1.26 | **42.23±1.40** |
| D5 | 17.89±5.89 | 73.78±2.17 | 68.19±1.96 | 72.21±1.07 | 67.94±1.52 | **76.61±1.64** | - | OOM | 67.06±1.90 | 73.26±1.50 | **79.97±0.43** |
| D6 | 31.43±2.15 | 69.91±0.65 | **71.60±0.94** | 69.70±1.15 | 30.90±1.01 | 59.92±0.97 | - | OOM | 7.09±0.24 | 47.79±1.04 | **81.68±0.60** |
| D7 | OOM | 31.74±1.25 | 47.70±1.47 | 49.06±1.02 | 51.57±2.15 | OOM | - | OOM | **54.22±0.64** | **55.24±1.12** | 53.90±3.93 |
| D8 | 79.12±5.89 | 58.54±2.17 | **82.96±1.96** | 82.60±1.07 | 75.79±1.52 | OOM | - | OOM | 76.18±1.90 | 80.87±1.50 | **84.41±3.00** |
| D9 | OOM | OOM | OOM | OOM | **23.38±0.91** | OOM | OOM | OOM | 22.20±0.50 | 21.97±1.08 | **40.63±3.58** |

**Figure 1: The clustering results of ACC on all datasets with different missing ratios.**

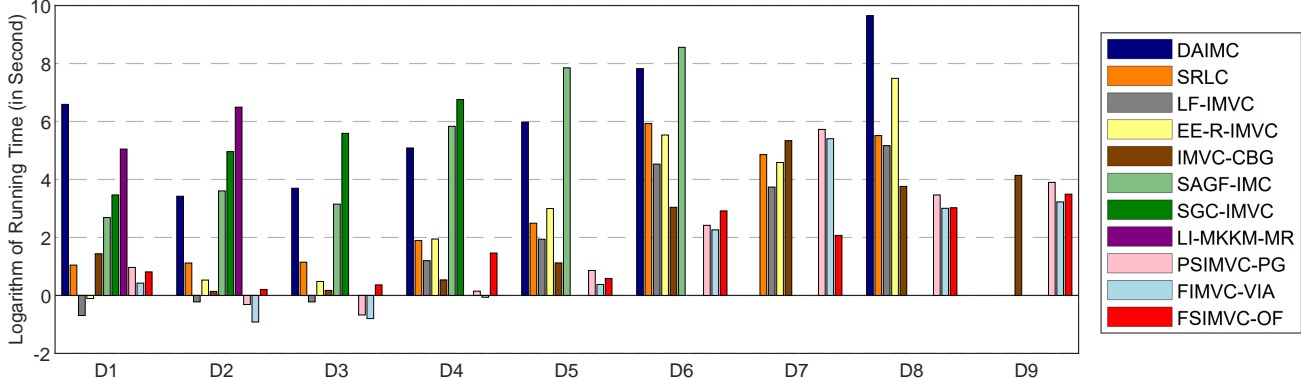

**Figure 2: Average running time comparison of different IMVC methods on nine datasets.**

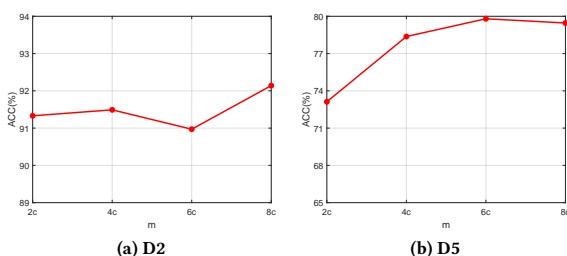

**Figure 3: ACC on different values of $m$ over two datasets.**

between the convergence process and the improvement in ACC values. Remarkably, across various datasets, our method consistently achieves convergence within approximately five iterations, underscoring the remarkable time efficiency of FSIMVC-OF.

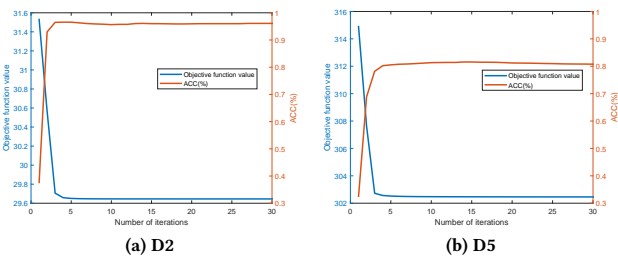

**Figure 4: The objective function value and ACC versus the number of iterations of the proposed method.**

### 5.7 Ablation Study

We evaluate the efficacy of adaptive graph filters in FSIMVC-OF through ablation experiments across various datasets. Comparing against two variants: 1) **FSIMVC**, where adaptive graph filters are removed, and clustering is based on bipartite graphs $\{\mathbf{B}^r\}_{r=1}^{v}$, and 2) **FSIMVC-GD**, applying view independent graph filter as Eq. (4) on each bipartite graph. The fixed parameter $\lambda = \frac{1}{2}$ aligns with the

**Table 3: Ablation result of FSIMVC-OF on five datasets.**

| Dataset | Method | FSIMVC | FSIMVC-GD | FSIMVC-OF |
|---------|--------|--------|-----------|-----------|
| D1 | ACC | 75.35±2.33 | 77.85±2.44 | **80.10±1.64** |
| | NMI | 82.76±1.91 | 84.71±1.33 | **86.20±0.87** |
| D2 | ACC | 91.91±1.00 | 92.40±1.23 | **92.67±1.30** |
| | NMI | 84.85±1.38 | 85.92±1.18 | **86.12±1.78** |
| D3 | ACC | 31.62±2.88 | 33.54±4.02 | **41.46±5.24** |
| | NMI | 7.17±2.36 | 8.94±4.01 | **19.18±4.44** |
| D8 | ACC | 79.56±3.94 | 79.71±3.90 | **81.14±4.24** |
| | NMI | 81.44±3.05 | 82.09±2.22 | **83.00±2.82** |
| D9 | ACC | 32.71±3.45 | 33.00±3.53 | **39.49±4.06** |
| | NMI | 10.53±2.27 | 10.92±2.20 | **17.18±2.93** |

primary experiment. Results in Table 3 highlight the full model's superior clustering efficacy, emphasizing the importance of adaptive graph filters in addressing incomplete multi-view clustering. Notably, FSIMVC-GD outperforms FSIMVC, showcasing the benefits of leveraging higher-order information. However, isolated graph filtering lacks adaptability, particularly evident in datasets D3 and D9, further supporting the superiority of FSIMVC-OF.

### 6 Conclusion

In conclusion, FSIMVC-OF leverages duality optimal graph filtering to enhance clustering in incomplete multi-view datasets with linear computational efficiency. We introduce a unified consensus clustering framework, supported by a fast optimization algorithm. Comparative experiments on nine datasets show the superiority of FSIMVC-OF over ten leading IMVC methods, underscoring its scalability and efficacy for large-scale IMVC tasks.

### 7 Acknowledgments

This work is supported in part by the National Natural Science Foundation of China grants 62376146, 62176001, the Natural Science Project of Anhui Provincial Education Department grants 2023AH030004.

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
