# OpenReview forum: "Fast and Scalable Incomplete Multi-View Clustering with Duality Optimal Graph Filtering"
_acmmm.org/ACMMM/2024/Conference — MM2024 Poster_

### Official Review · Reviewer_C5jN · 2024-05-23

**Rating:** 4
**Confidence:** 2

**Summary:**

The paper proposes a fast and scalable incomplete multi-view clustering with duality optimal graph filtering, which refines the clustering-friendly structure of the bipartite graph by learning an optimal filter within a consensus clustering framework. It leverages optimal graph filter learning on the sample side to encapsulate higher-order interactions, guiding the enhancement of clustering-friendly structures for the first-order bipartite graphs.

**Strengths:**

1. The paper exhibits a well-defined structure with cogent and logical writing throughout.
2. The authors have provided access to the code to facilitate reproducibility, ensuring it is meticulously organized.
3. The experimental section of the study is extensive, benchmarking the proposed approach against a range of state-of-the-art rivals on multiple datasets, notably including those of a large-scale dataset.

**Limitations:**

I would like to see ablation experiments on all datasets.

**Suitability:**

3

---

### Official Review · Reviewer_LRHD · 2024-05-23

**Rating:** 4
**Confidence:** 3

**Summary:**

The paper presents a novel approach to addressing the challenge of incomplete multi-view clustering in multimedia data analysis. The authors identify the limitations of existing graph learning-based IMVC methods, particularly their inability to accurately classify samples due to data incompleteness and their high computational demands. To overcome these issues, the paper introduces a new method that employs duality optimal graph filtering within a consensus clustering framework. This approach refines the bipartite graph structure by learning an optimal filter, which is applied to the anchor side for computational efficiency. The authors propose an alternative optimization algorithm with linear complexity, which is well-suited for large-scale tasks. Extensive experiments demonstrate the effectiveness of the proposed method.

**Strengths:**

1.	The paper introduces a novel method that tackles the problem of incomplete data in multi-view clustering by using duality optimal graph filtering.
2.	The authors have designed an optimization algorithm with linear complexity, making the approach highly scalable and suitable for large datasets.
3.	The authors offer theoretical underpinning for their approach.

**Limitations:**

1.	While the paper claims linear complexity, the actual implementation and optimization of the proposed method might be complex.
2.	Why does the parameter m have a limited effect on clustering performance?

**Suitability:**

3

---

### Official Review · Reviewer_eFMe · 2024-05-24

**Rating:** 5
**Confidence:** 3

**Summary:**

This paper proposes a novel method aimed at exploring multi-view missing data clusters based on more advanced graph signals. Initially, it constructs the first-order correlation between samples and anchor points based on the local information of the complete samples. By leveraging the equivalence between the sample-side filtering graph and the anchor-side filtering, it obtains graph signals containing more information. This leads to consistent clustering indicators across multiple views.

**Strengths:**

1. The experimental results are almost state-of-the-art, demonstrating good time complexity and convergence.
2. The method also exhibits relatively stable performance under different missing rate conditions.

**Limitations:**

However, there are several points that require further clarification and improvement:
1. The authors repeatedly emphasize that the "prevalent first-order affinity graph often misclassifies out-neighborhood intra-cluster and in-neighbor inter-cluster samples," but there is no relevant description explaining how the proposed method addresses this challenge. Including some experiments with toy data might help readers better understand how the method overcomes this specific difficulty.
2. The concept of graph filtering on the sample side and anchor side is intriguing. However, it is not clear how the graph filters capture higher-order information.
3. In the final optimization step, all known information from the views is aggregated, and a probabilistic constraint is applied. It raises the question of why not directly partition the clusters based on the probability sizes for the c clusters in Z. Why is it necessary to perform SVD decomposition followed by k-means clustering to obtain the clustering results?
4. In section 5.2, the authors mention comparison methods based on graph filtering. However, I could not find these methods in the references. Clarifying the differences between the proposed method and other graph filtering-based methods would strengthen the paper.
5. There are also minor errors in the notation. For example, in line 299, it should be $\mathbb{R}$, and in line 324, $\{P^r\}_{r=1}^m$ should be $v$ not $m$, be consistent with the previous notation.

**Suitability:**

3

---

### Meta-Review · Area_Chair_gGQC · 2024-07-04

**Recommendation:** Accept (Poster)
**Confidence:** 5

**Metareview:**

The paper received 3 reviews, recommending weak accept, borderline accept, borderline accept
The main strengths highlighted by the reviewers are the extensive evaluation of the proposed method and the suitability of the contribution to the conference.
The weaknesses seem to have been addressed in the rebuttal document – also for those reviewers that did not comment of the rebuttal in their reviews
Taking into account  the above, I recommend accepting the  paper for poster presentation.